# Study on Reradiation Interference Characteristics of Steel Towers in Transmission Lines

**Li Huang** [1,2,*], **Bo Tang** [1], **Xingfa Liu** [2] **and Jianben Liu** [2]

1   College of Electrical Engineering and New Energy, China Three Gorges University, Yichang 443002, China
2   State Key Laboratory of Power Grid Environmental Protection, China Electric Power Research Institute, Wuhan 430074, China
*   Correspondence: huangli@ctgu.edu.cn

**Abstract:** With the development of ultrahigh-voltage transmission lines in China, the reradiation interference caused by the steel tower used in ultrahigh-voltage transmission becomes increasingly aggravating for nearby radio stations. In this paper, using the multilevel fast multipole algorithm, the reradiation interference of an ultrahigh-voltage transmission steel tower is investigated. Additionally, the reradiation interference characteristics of a transmission steel tower were investigated with various frequencies and azimuth angles of incidence wave. The results show that the frequency and the azimuth angle of incident wave are the impact factors for the reradiation interference of steel towers. It is better to use a detailed angle steel model for the truss structure of steel towers at high frequency, while a simplified structural model can be used at low frequency. Additionally, the diffraction at the edge of the angle steel has a great influence on the reradiation interference, particularly at high frequencies.

**Keywords:** reradiation interference; steel tower; transmission line

## 1. Introduction

With the implementation of "the new infrastructure" in China [1], UHV (ultrahigh-voltage) transmission lines have been confirmed as one of the infrastructure constructions. Additionally, UHV transmission lines have progressively increased over the past few years [2]. However, the large metal structure in UHV transmission lines may be effective reradiators of radio waves, which create an undesirable distortion in the radiation pattern of radio stations in the vicinity. From a point of view of radio wave propagation, the electromagnetic wave from a radio station induces electric currents into the metal structure in the transmission line, and these induced currents radiate their own electromagnetic waves, which may alter the effective far-field pattern of radio stations [3,4]. With the spread of urban populations, the reradiation interference from UHV transmission lines becomes more serious [5].

Compared with the complex and expensive experimental study on the reradiation problem of transmission steel towers, the simulation research based on numerical calculation technology has been widely promising. Trueman and Kubina introduced the "uniform stem" tower model in the early research and used the "tapered stem" tower model with full cross-arm detail in their subsequent study [6–8]. Zhao and Gan used a thin wire instead of the angle steel in a steel tower to generate the wire model and analyze the reradiation about the loop consisting of two transmission towers, overhead ground wires and the ground image [9,10]. Tang conducted an analysis of the wire model and the surface model of steel towers in the prediction of the reradiation interference calculation and gave some suggestions about the application of each model [11,12].

In the traditional research on the reradiation interference of transmission lines, the induced current is distributed along a loop composed of the ground wire, the steel tower and the ground [11–13]. However, in UHV transmission lines, the reradiation interference

from the steel tower alone accounts for the main part due to the segmented insulation between steel towers and grounding wires. Additionally, the reradiation interference of the steel tower needs to be analyzed separately. Furthermore, the established model should describe the real steel tower in more detail; it is difficult to build a suitable steel tower model in the calculation. The lattice structure and the angle steel materials of steel towers increase the complexity of the model. Although the MOM (method of moment) can obtain high-accuracy results [14–16], the memory requirement and calculation time of the MOM increases dramatically with the frequency of incident waves and the complexity of the model, which limits the application of the MOM in the reradiation interference of the transmission line. In order to obtain the reradiation interference at high frequency, several asymptotic high-frequency numerical methods, such as PO (physical optics) [17,18], UTD (uniform theory of diffraction) [19,20] and so on, have been introduced. Considering the radio wave is assumed to be the light wave in asymptotic high-frequency numerical methods, these methods have poor computational precision in the reradiation interference calculation of the transmission line.

In this paper, the definition of reradiation interference of steel towers is given first. Then, the MLFMA (multilevel fast multipole algorithm) based on the MOM numerical is introduced to calculate the reradiation of a steel tower in a UHV transmission line. Additionally, modeling methods of steel towers using parameters are presented. At last, the reradiation interference characteristics of a steel tower and its relation to the feature of incidence waves are investigated.

## 2. Method of Reradiation Interference Calculation

### 2.1. Reradiation Interference

According to the definition of reradiation interference, the influence of reradiation interference generated by a steel tower can be written below [21]

$$RRI = 20lg\frac{|E_r + E_i|}{|E_i|},\qquad(1)$$

where $RRI$ is the value of the reradiation interference due to the steel tower at the observation point, $E_r$ is the reradiation electric field at the observation point generated by the induced current on the surface of the steel tower and $E_i$ is the incident electric field at the same observation point due to the incident wave from the radio station. Considering the electric field $E_i$ can be known when the radio station is determinate, the calculation of the reradiation electric field $E_r$ becomes the key issue in the study.

### 2.2. Numerical Calculation of Reradiation Electrical Field

Using the theorem of electromagnetic field, at the observation point $r$, the reradiation electric field $E_r(r)$ generated by induced current density $J(r')$ and charge density $\rho(r')$ can be written as

$$E_r(r) = j\omega\mu\int_V g(r,r')J(r')dr' - \nabla\frac{1}{\epsilon}\int_V g(r,r')J(r')dr'\qquad(2)$$

where, $g(r,r')$ is Green's function and $g(r,r') = \frac{e^{-jk|r-r'|}}{4\pi|r-r'|}$, $\omega$ is angular frequency, $\mu$ is permeability and $\epsilon$ is permittivity. In this paper, the material of the steel tower is assumed to be a perfect conductor, and the tangential component of the electric field on the surface of the steel tower should be equal to zero. Thus, using (2), the relationship between the incident electromagnetic wave $E_i(r)$ and the induced current density $J(r')$ can be obtained as below

$$-t \cdot E_i(r) = j\omega\mu t \cdot \int_S g(r,r')J(r')dS' - \frac{1}{j\omega\epsilon}t \cdot \int_S g(r,r')\nabla' \cdot J(r')dS'\qquad(3)$$

where $t$ is the tangential unit vector. Using the the MOM to solve the equation [22], the surface of the steel tower will be subdivided into many triangular patches, and the RWG

(RAO, WILTON and GLISSON) basis function $f_n(r)$ will be employed to solve the unknown current on each pair of patches [23]. The basis function $f_n(r)$ is defined on the patches $T_n^-$ and $T_n^+$, as shown in Figure 1.

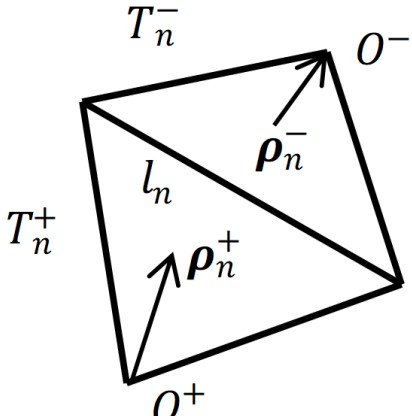

**Figure 1.** The pair of triangular patches in the RWG base function.

The basis function $f_n(r)$ can be written as

$$
f_n(r) = \begin{cases}
\frac{l_n}{2A_n^+}\rho_n^+ = \frac{l_n}{2A_n^+}(\rho - \rho_{no}^+), & \rho \in T_n^+ \\
-\frac{l_n}{2A_n^-}\rho_n^- = -\frac{l_n}{2A_n^-}(\rho - \rho_{no}^-), & \rho \in T_n^- \\
0, & other
\end{cases}
\tag{4}
$$

where $T_n^\pm$ are the triangular patches, $l_n$ is the inner edge of $T_n^\pm$, $A_n^\pm$ are the areas of $T_n^\pm$ and $\rho_n^\pm$ are the vector from the vertex point $O^\pm$ to the point on the patches, separately. Additionally, the surface-induced current density is expanded with the basis function as

$$
J = \sum_{n=1}^{N} I_n f_n
\tag{5}
$$

where $I_n$ is the weighting coefficients of the basis function on the $n$th pair of triangular patches. Substituting (4) and (5) into (3), and using the Galerkin testing method in the MOM, (3) can be transformed into a linear equations system as below

$$
\sum_{n=1}^{N} Z_{mn}I_n = V_m, \quad m = 1, 2, \dots, N
\tag{6}
$$

where

$$
Z_{mn} = j\omega\mu \int_S \int_{S'} [f_m(r) \cdot f_n(r') - \frac{1}{\omega^2\mu\epsilon}\nabla \cdot f_m(r)\nabla \cdot f_n(r')]g(r,r')dSdS'
\tag{7}
$$

and

$$
V_m = \int_S E_i(r)f_m(r)dS
\tag{8}
$$

The value of $Z_{mn}$ can be considered for the interactions between the $m$th pair of triangular patches and the $n$th pair of triangular patches. The weighting coefficient $I_n$ can be calculated by solving (6), and the induced current density $J$ can be obtained by (5). Then, the reradiation electric field $E_r$ can be found according to (2).

In order to reduce the memory requirement and computational complexity, the MLFMA is employed to solve (6). The steel tower is first divided into groups. Assume the $m$th pair of triangular patches is located in group $i$, and the $n$th pair of triangular patches is located in group $j$. Then, the interactions among the pairs of triangular patches in the steel

tower are classified into the near-field interaction and the far-field interaction due to the distance between the two groups $i$ and $j$. For the near-field interaction, $Z_{mn}$ is evaluated using (7) directly. For the far-field interaction, the distance between the pairs of triangular patches can be expressed as

$$\boldsymbol{r} - \boldsymbol{r}' = \boldsymbol{r} - \boldsymbol{a} + \boldsymbol{a} - \boldsymbol{b} + \boldsymbol{b} - \boldsymbol{r}' = \boldsymbol{r}_{ra} + \boldsymbol{r}_{ab} - \boldsymbol{r}_{r'b} \tag{9}$$

where the subscripts $a$ and $b$ represent the center of groups $i$ and $j$, and $\boldsymbol{r}_{ab}$ is the spatial vector from the center of group $i$ to the center of group $j$. Considering $|\boldsymbol{r}_{ab}| > |\boldsymbol{r}_{ra} - \boldsymbol{r}_{r'b}|$, using the Taylor series expansion, Green's function can be approximated in the far field as

$$\frac{e^{-jk|\boldsymbol{r}-\boldsymbol{r}'|}}{4\pi|\boldsymbol{r}-\boldsymbol{r}'|} = \frac{e^{-jkr_{ab}}}{4\pi|\boldsymbol{r}_{ab}|} \cdot e^{-jkk_0 \cdot \boldsymbol{r}_{ra}} \cdot e^{-jkk_0 \cdot \boldsymbol{r}_{r'b}} \tag{10}$$

where $k_0 = \frac{r_{ab}}{|r_{ab}|}$, $k_0$ is the direction from the group $i$ to the group $j$. Hence, the far-field interaction can be first translated from the pair of triangular patches within a group into a single center using the addition theorem, then received by the different group centers and at last redistributed to the pair of triangular patches belonging to these groups, similarly. Considering the far-field interaction can be grouped together before the integral, the matrix-vector product can be accelerated by the MLFMA.

### 3. Analysis Model of Steel Towers

The structure and shape of steel towers determine the distribution of induced current and impact on the reradiation interference from steel towers. Thus, it is better to generate the simulation model, which is very close to a realistic steel tower, in the analysis of the reradiation interference characteristics.

However, a steel tower is a massive latticed metal structure comprised of a large number of angle steels. According to the bearing internal force and function in steel towers, the angle steels may have different lengths, thickness and leg widths. Additionally, it raises not only the complexity of the simulation model but also the number of elements in the field calculation.

Considering the thickness of the angle steel is sufficiently short, comparable to the wavelength of the incident electromagnetic wave, the induced current on the inner surface and the outer surface of the angle steel will be thought to follow the same pattern as the incident electromagnetic wave. Additionally, the thickness of the angle steel can be neglected in the calculation, and the angle steel is simulated with the two adjacent conductor surfaces, as shown in Figure 2.

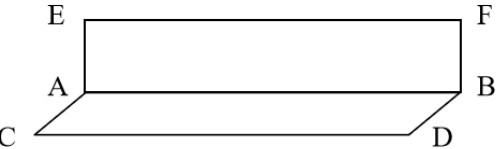

**Figure 2.** The surface model of angle steel in the reradiation interference calculation.

The angle steel in the calculation model can consist of two rectangles, as shown in Figure 2, and be determined by the points $\boldsymbol{A}$, $\boldsymbol{B}$ and $\boldsymbol{C}$. Then, according to the relationship of the vector, the remaining points in the model can obtained as below

$$\begin{cases} \boldsymbol{D} = \boldsymbol{B} + \boldsymbol{C} - \boldsymbol{A} \\ \boldsymbol{E} = (\boldsymbol{C} - \boldsymbol{A}) \times (\boldsymbol{B} - \boldsymbol{A}) \frac{||\boldsymbol{C}-\boldsymbol{A}||}{||(\boldsymbol{C}-\boldsymbol{A})\times(\boldsymbol{B}-\boldsymbol{A})||} \\ \boldsymbol{F} = \boldsymbol{B} + \boldsymbol{E} - \boldsymbol{A} \end{cases} \tag{11}$$

where $\boldsymbol{A}$, $\boldsymbol{B}$ and $\boldsymbol{C}$ are the vertex vectors of A, B and C, respectively. According to the assembly drawing of a steel tower, the vertex vector list of each angle steel can be recorded

in sequence. Thence, applying the script language, the analysis model of the steel tower can be created automatically. In order to eliminate the non-coplanar phenomenon between the surfaces of adjacent angle steel, it is better to use float or double data type during modeling.

Considering that steel towers have a low height and good stability and are widely used in transmission lines, a typical UHV transmission steel tower was modeled and analyzed in the paper. The assembly drawing of the steel tower is shown in Figure 3.

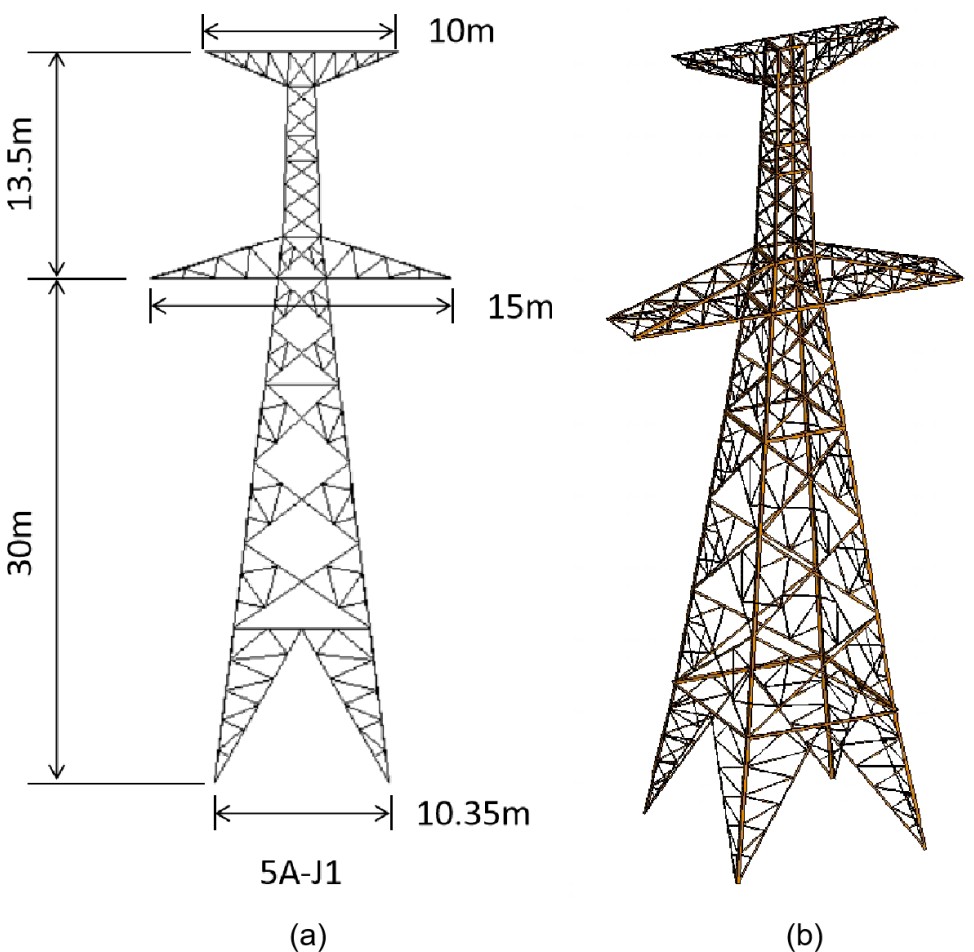

(a)  (b)

**Figure 3.** A assembly drawing of the steel tower in the model. (**a**) Assembly drawing. (**b**) Angle steel in the model.

## 4. Case Study

In the case study, the center of the steel tower model was placed at the origin of the coordinate, as shown in Figure 4. The conductor and the ground wire in the transmission line were negligible because of the insulation from the steel tower. The incident wave uses a parallel plane wave with a 90° azimuth angle and VV polarization. The observation point is placed on the $x$-axis to receive the reradiation interference of the transmission line.

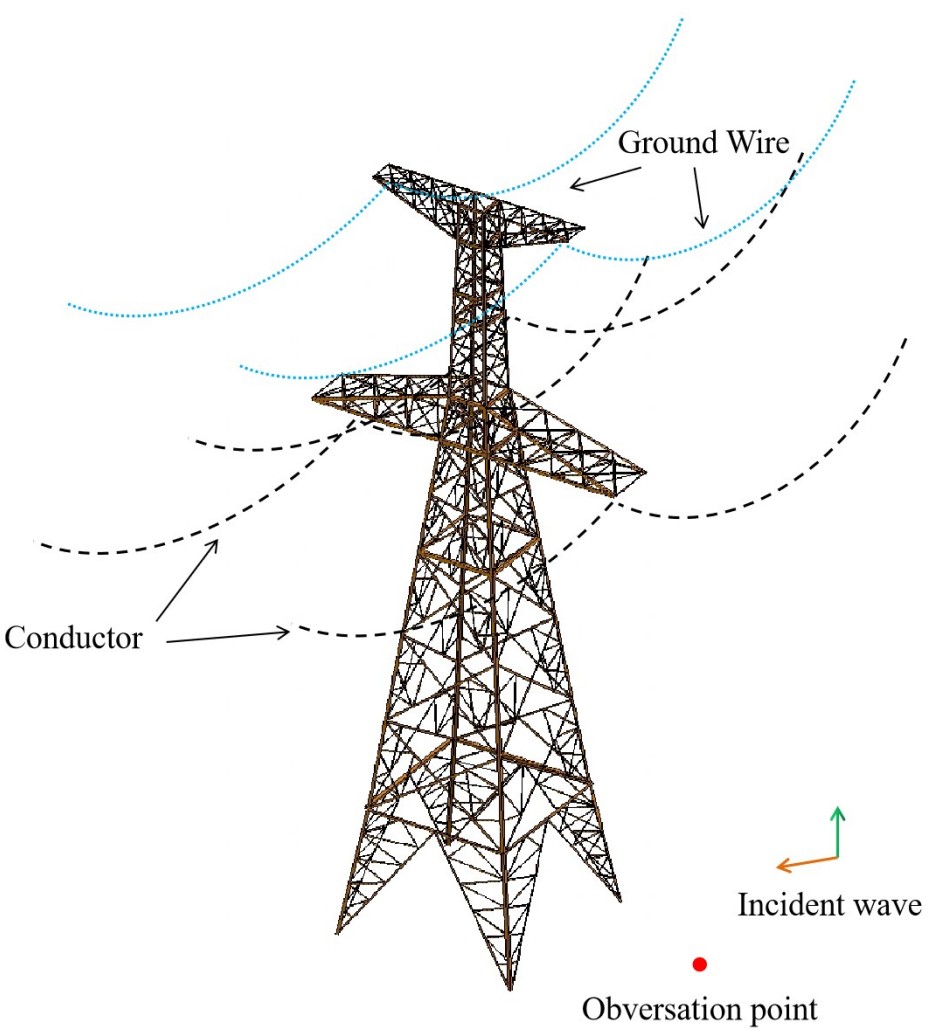

**Figure 4.** The calculation model of the steel tower.

Using the method mentioned above, the surfaces of the steel tower model were segmented into several triangles with patch length of about $\lambda/12$. Additionally, the ground is assumed to be an infinite perfect conductor plane in the analysis for simplicity.

### 4.1. Calculation Method Verification

In order to verify the effectiveness of MLFMA in the reradiation interference calculation, a typical UHV transmission steel tower was established according to the above method. The frequency of the incident wave is 100 MHz, and the observation azimuth angle is from 0° to 90°. The calculation results of the method proposed in the paper are compared with the exact numerical algorithm results based on the MOM and the asymptotic high-frequency numerical algorithm results based on the PO.

As shown in Figure 5, the calculation results of the MOM and MLFMA are in good agreement at different azimuth angles, and the difference of the two results is fewer than 0.001 dB. However, the maximum deviation of calculation results of PO is close to 0.054 dB. Furthermore, the calculation time of the MOM is about 25.77 times that of the MLFMA, and the memory requirement of the MOM is about 4.88 times that of the MLFMA.

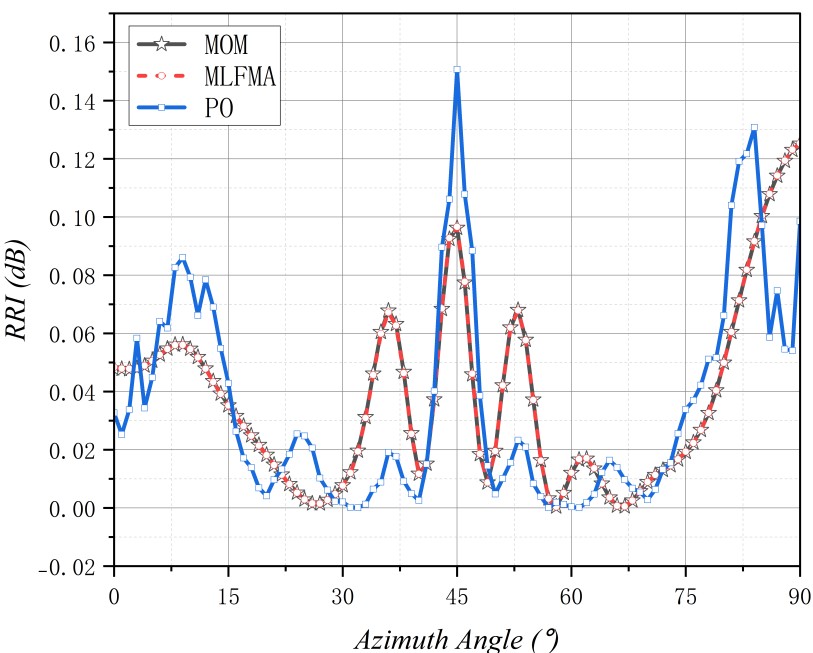

**Figure 5.** The comparison of the reradiation interference at 100 MHz.

### 4.2. Influence of the Frequency of Incident Waves

Due to their complex latticed metal structure, it is difficult to predict the resonant frequency of steel towers. However, steel towers behave as vertically grounded conductors, and the maximum interference value is approximated to occur at "$\lambda/4$ resonance frequency". Considering the height of a steel tower is 43.5 m, the resonance wavelength can be calculated as 174 m, 4 times the height of a steel tower. Furthermore, the resonant frequency of a steel tower can be obtained as 1.724 MHz.

In the calculation model, the value of the frequency of the incidence wave ranged from 0.1 MHz to 100 MHz; the reradiation interference is calculated and shown in Figure 6. It is evident that the reradiation interference is much worse at some certain frequencies, which lie closer to the resonant frequency of the steel tower. The actual resonant frequency may be at 1.4895 MHz and is lower than the calculation result of 1.724 MHz because the cross-arms of the steel tower change the distribution of the induced current. Additionally, the induced horizontal current along the top cross-arm and the bottom cross-arm has an antiphase to the mirror current in the ground. Although these currents do not produce radiation, the height of the steel tower was increased equivalently. Additionally, the resonant frequency of the steel tower is decreased correspondingly.

### 4.3. Influence of the Azimuth Angle of Incident Waves

Considering the structural profile of steel towers are not the same in different directions, the reradiation interference can be changed by the azimuth angle of the incident wave. In the analysis, the incident wave with azimuth angles 0°, 30°, 60° and 90° was employed in the model, and the results are shown in Figure 7.

When the frequency is lower than 6.3 MHz, it can be found that there was almost no difference when the azimuth angle of incident wave is changed from 0° to 90°. In this case, the model of the steel tower may be simplified using a symmetrical geometry such as cylinders, cubes and so on. However, as the frequency of the incident wave increases, the difference between the azimuth angles of the incident wave become obvious. Additionally, the model should be considered in more detail, such as the position of auxiliary material, the leg length of the angle steel and so on.

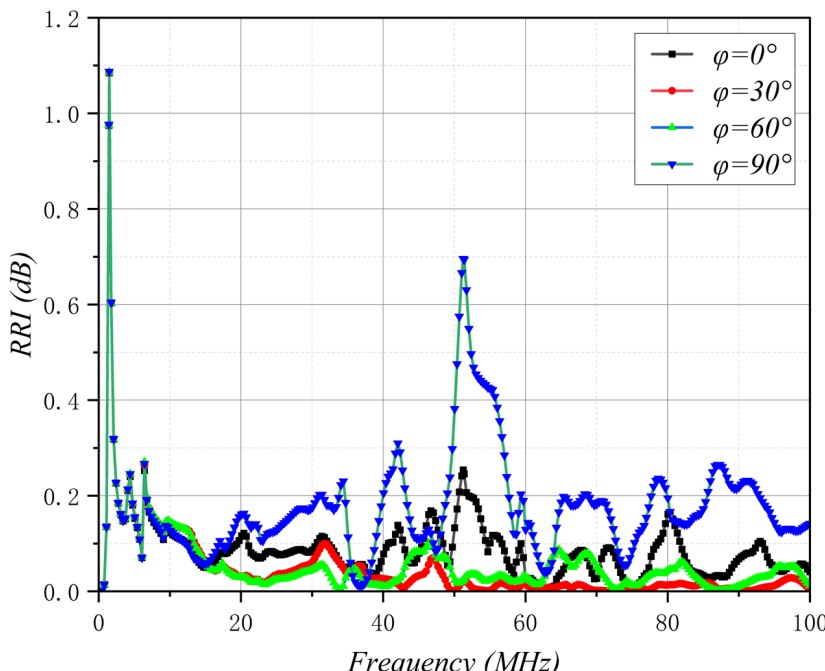

**Figure 6.** The reradiation interference of the steel tower varies with the frequency of the incident wave at azimuth angles 0°, 30°, 60° and 90°.

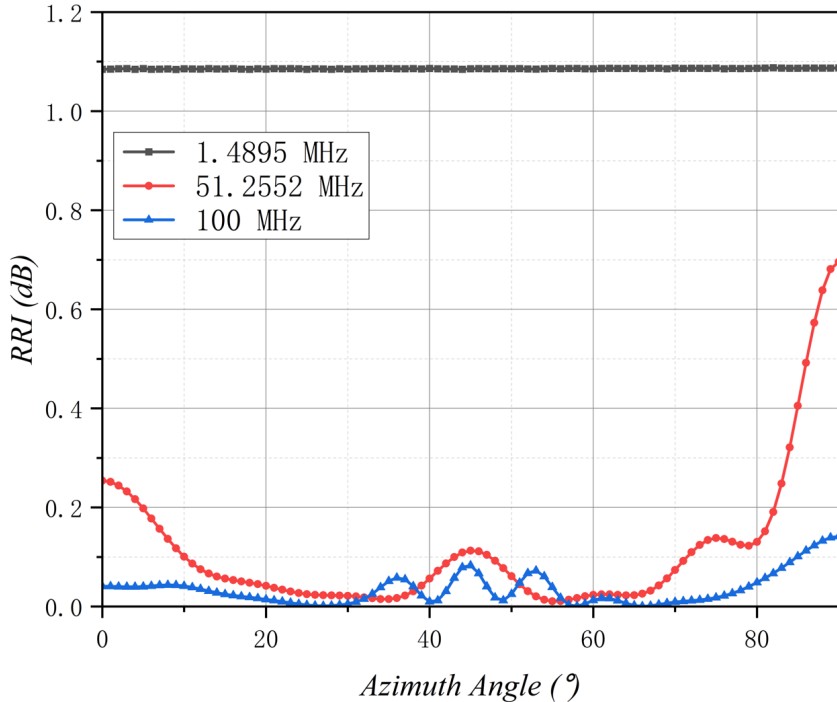

**Figure 7.** The reradiation interference of the steel tower varies with the azimuth angles at different frequencies.

When the frequency is 1.4895 MHz, the reradiation interference of steel towers is almost the same at different azimuth angles. The influence of steel towers' shape is small, and the model of the steel tower can be approximated by a cylinder. As the frequency increases to 51.2552 MHz, the influence of the steel tower's shape is revealed. There are several local maximum values of reradiation interference at azimuth angles 45°, 75°, etc. This means that the edge diffraction effect of the steel tower increases, and the phenomenon

of edge diffraction effect becomes more obvious at 100 MHz. Therefore, in the high frequencies, the reradiation interference analysis of steel towers requires the use of a model with more truss detail. At the same time, the magnitude of the interference also decreases as the frequency increases.

### 4.4. Influence of the Height of Steel Towers

In order to ensure the sufficient safety distance between the conductor of the transmission line and the ground, the height of a steel tower will increase or decrease according to the undulations of the terrain. In the analysis, the height of the steel tower is changed from 41.5 m to 45.5 m, and its reradiation interference was calculated using the previous method. The calculation results are shown in Figure 8.

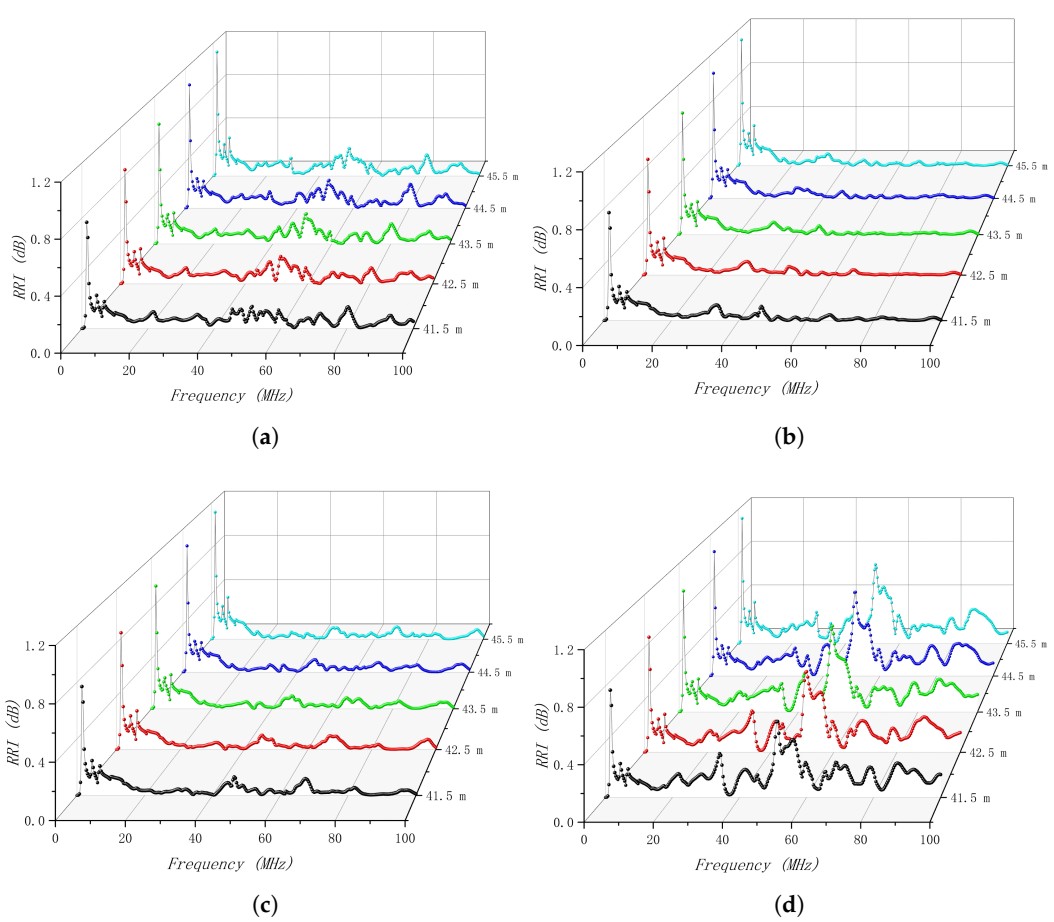

**Figure 8.** The reradiation interference of the steel tower varies with the frequencies at the different azimuth angles. (**a**) Azimuth angle is 0°. (**b**) Azimuth angle is 30°. (**c**) Azimuth angle is 60°. (**d**) Azimuth angle is 90°.

As the frequency changes, the height of a steel tower has less influence on the amplitude of the reradiation interference. A change in the height of a steel tower will bring about changes in the resonant frequency, as mentioned in the previous section. In addition, the increase in the height of a steel tower will equivalently increase the reflection area of a steel tower to electromagnetic waves, which will cause an increase in the reradiation interference, as shown in Figure 9. It can also be found that the reradiation interference of a steel tower is large at low frequencies. Additionally, the increase in the height of a steel tower has the potential to cause a reduction in the reradiation interference due to the massive latticed metal structure and angle steels in the steel tower.

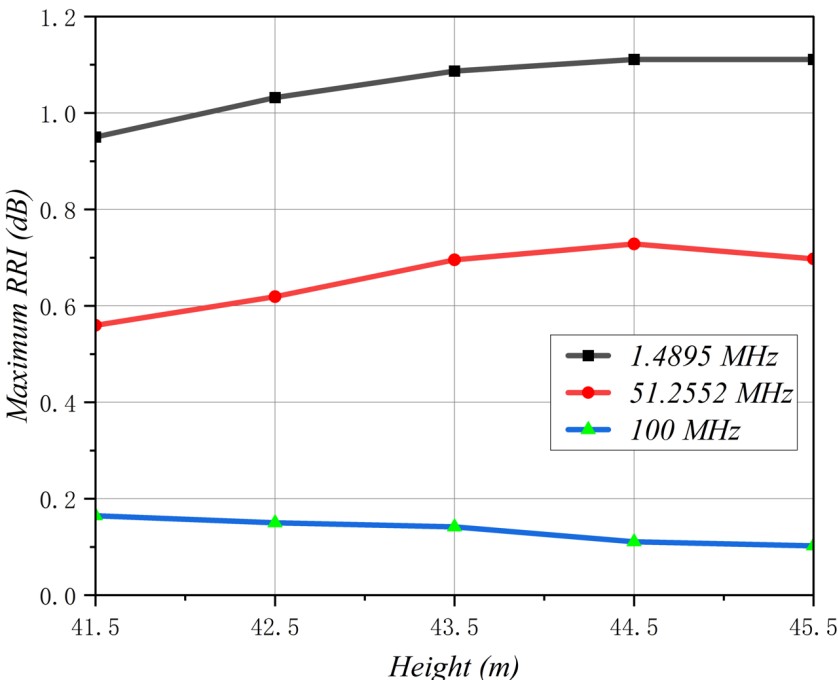

**Figure 9.** The reradiation interference of the steel tower varies with height at the different frequencies.

### 4.5. Influence of the Cross-Arm of Steel Towers

Considering that the cross-arms changes the distribution of the induced current on the surface of a steel tower, the magnitude of the reradiation interference of a steel tower will also be affected. In this study, the lengths of the upper cross-arm and lower cross-arm were changed, respectively, and the reradiation interference was calculated and shown in Figure 10.

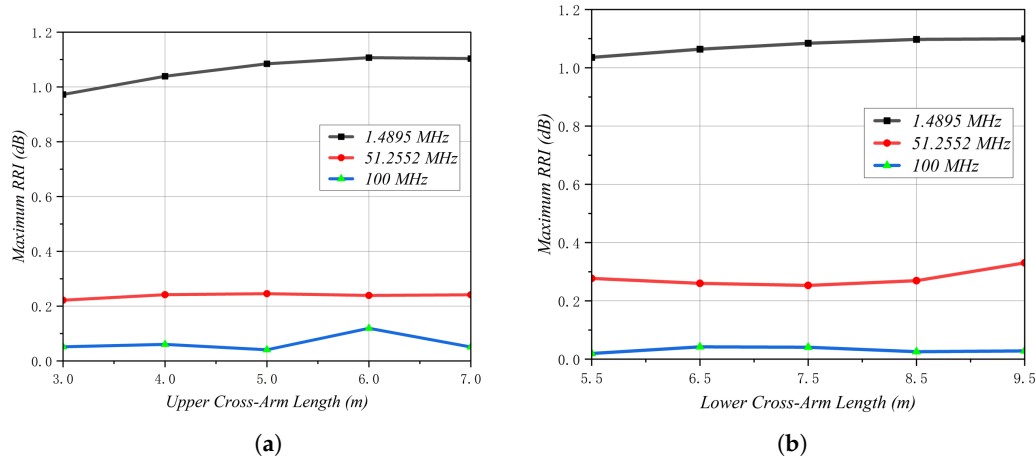

**Figure 10.** The reradiation interference of a steel tower varies with the length of cross-arm at the different frequencies. (**a**) Upper cross-arm. (**b**) Lower cross-arm.

Similar to changes caused by the height of a steel tower, the reradiation interference is less affected by the cross-arm length. When the frequencies are 1.4895 MHz, 51.2552 MHz and 100 MHz, and the length of the cross-arm changes less than 5 m, the fluctuation of reradiation interference is within 0.1 dB.

### 4.6. Influence of the Distance Away from Steel Towers

Assuming the protected distance of reradiation interference between the UHV transmission line and the radio station is 2 km, the magnitude of reradiation interference at the

observation points, which are from 0.4 km to 2 km on the *x*-axis, were calculated, and the results are shown in Figure 11.

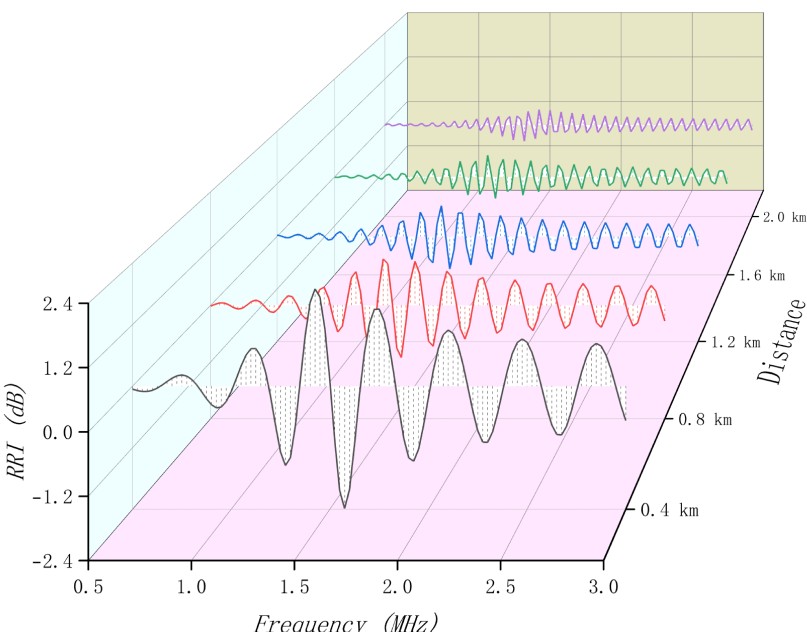

**Figure 11.** The reradiation interference at different observation points along the *x*-axis.

The reradiation interference of transmissions of steel towers was a decay oscillation curve and decreased with the frequency of the incident wave increasing. It is difficult to have a determined value to describe the reradiation interference at different frequencies as a function of distance. Thus, the maximum value of the magnitude of the reradiation interference decreased with the distance away from the steel tower, as shown in Figure 12. It can be found that the magnitude of reradiation interference is already less than 0.4 dB at 2 km and is appropriate to keep the distance between the UHV transmission steel tower and the radio station at more than 2 km in the design.

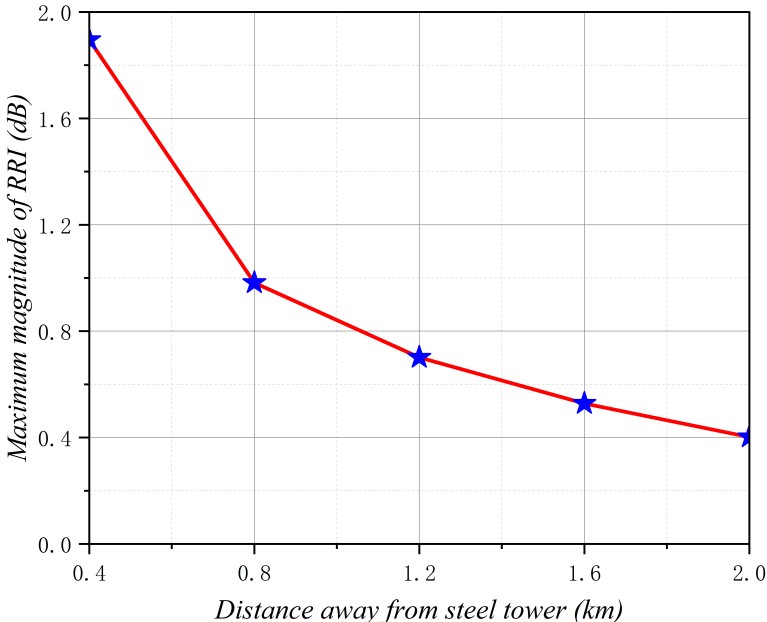

**Figure 12.** The maximum magnitude of the reradiation interference with the distance away from the steel tower.

## 5. Conclusions

In this paper, a numerical calculation method for the reradiation interference of steel towers was described. Additionally, the model of the steel tower in the UHV transmission line was constructed to establish reradiation interference characteristics. The method can not only establish the surface model of steel towers conveniently but also obtain calculation results similar to the computational precision of the MOM. Furthermore, the calculation time and the memory requirement of the method are much less than that of the MOM method. It is efficient and convenient to apply the method to analyze the reradiation interference of steel towers and aid in the design and construction of transmission lines, as well as radio station locations. As shown in the case study, both the height and the length of a cross-arm have little effect on the reradiation interference in steel towers. When the azimuth angle is at 45° and its multiples, the reradiation interference of a steel tower may appear at larger values. Therefore, the analysis of the reradiation interference of steel towers in UHV transmission lines should pay attention to the influence of frequency, especially the resonance frequency, and the direction to the steel tower. Furthermore, the calculation model should have more details on high frequency for reliable analysis of transmission steel towers.

**Author Contributions:** Conceptualization, L.H. and B.T.; methodology, L.H.; software, L.H.; validation, X.L. and J.L.; formal analysis, L.H.; investigation, X.L. and J.L.; resources, B.T.; data curation, L.H.; writing—original draft preparation, L.H.; writing—review and editing, B.T. All authors have read and agreed to the published version of the manuscript.

**Funding:** This research was supported by the Open Fund of State Key Laboratory of Power Grid Environmental Protection grant number GYW51202001549.

**Institutional Review Board Statement:** Not applicable.

**Informed Consent Statement:** Not applicable.

**Data Availability Statement:** Not applicable.

**Acknowledgments:** This paper is supported by Open Fund of State Key Laboratory of Power Grid Environmental Protection (No. GYW51202001549).

**Conflicts of Interest:** The authors declare no conflict of interest.

## Abbreviations

The following abbreviations are used in this manuscript:

| | |
|---|---|
| UHV | Ultrahigh Voltage |
| MLFMA | Multilevel Fast Multipole Algorithm |
| MOM | Method of Moment |
| PO | Physical optics |
| SBR | Shooting and bouncing rays |
| UTD | Uniform theory of diffraction |
| RWG | RAO, WILTON and GLISSON |

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
