# Peer review of "Study on Reradiation Interference Characteristics of Steel Towers in Transmission Lines"

_information, doi:10.3390/info13110521_

Round 1
Reviewer 1 Report (Previous Reviewer 2)
In the reviewed manuscript the radiation interference of the ultra-high voltage transmission steel tower was analyzed. The radiation interference characteristics of transmission steel tower had been investigated with various frequencies and azimuth angles of incidence wave.
The paper is quite interesting but I have few remarks:
· The paper includes only numerical analysis of the problem.
· In the paper there is no comparative analysis with other methods.
· The paper is purely computational, without measurements. The problem discussed in this paper can be calculated with the use of known numerical packages.
· Figures 8 and 11 are illegible.
· In section Conclusions, should be mentioned advantages and disadvantages of the proposed solution.
· The novelty of the paper must be more clearly demonstrated.
· References section is very poor and should be extended.
Author Response
Thank you for your review, please see the attachment.

Reviewer 2 Report (Previous Reviewer 1)
I have reviewed the results and they seem logical to me, since the reradiation dependes on the geometry of the tower and the wavelength of the incident signals, so the problems is more complex at high frequency.
I consider that is accepted specifying the following points.
1-source location or what type of source (eg. isotropic)
2- appropiated wording of the paragraph of line 158-162
Author Response
Thank you for your review, please see the attachment.

Round 2
Reviewer 1 Report (Previous Reviewer 2)
The references section should be more extensive.
Author Response
Response to Reviewer 1 Comments
Point 1: The references section should be more extensive.
Response 1: Thank you very much for your constructive suggestions. And we have added some relevant references to the manuscript.
This manuscript is a resubmission of an earlier submission. The following is a list of the peer review reports and author responses from that submission.
Round 1
Reviewer 1 Report
As you know, the topic you addressing is not new and the part that you present as novel is the use of the mulilevel fast multipole applying the MoM, which is really a different analysis methdology . The resulta are difficult to corroborate even though some of them are logical from my point of view.
In the 64 line there is a mistake
If you make a description of how the results are obtained and how the UHV transmission lines affects the RRI, their work woulbe much better
Reviewer 2 Report
In the reviewed manuscript a numerical calculation method for the radiation interference of steel tower was proposed.
The paper is quite interesting but I have few remarks:
· The paper includes only numerical analysis of the problem.
· In the paper there is no comparative analysis with other methods.
· The paper is purely computational, without measurements.
· The Conclusion section is very poor.
· The novelty of the paper must be more clearly demonstrated.
· In section Conclusions, should be mentioned advantages and disadventages of the proposed method.
· Please supplement the paper with further options for modification/extension of the proposed method.
· References section is very poor and should be extended.
· ect.
·
Round 2
Reviewer 1 Report
As mentioned, there are publication about of the topic. The most similar is the reference [9] (same author) the difference is that now presented Multi-Level Fast Multipole Algorithm (MLFMA) and the results are compared with MOM using the model of steel angle.
The limits of expression (1) are 0 dB to 6 dB, but the incident field is not specified I assume that is normalized.
The tower structure is made for lattice of steel angle and the dimension define the spectrum of the radiation frequency. This characteristic is important because justified the resonance and it is not mentioned.
Figures 5-6 not present the MOM and It cannot be compared with the MLFMA. If SBR has not the same condition of the other methods, there is no point in showing up
If the electromagnetic waves incident to tower structure produces current density that generate radiated field, then the behavior is of an antenna and this will resonate at multiples of wavelengths with respect to the incident frequencies, since the structure of the lattice is uniform.
The effect of the heights of the towers depends on the position of the source which is not specified, which are not uniform. In order to reach adequate conclusions, more work is required.
The case of the influence of the Cross-arm of Steel Tower analysis is logical because the wavelength is different for each frequency. The results presented are not spectacular.
Reviewer 2 Report
References section is very poor.